

# The role of improved soil moisture for the characteristics of surface energy fluxes in the ECMWF reanalyses

Wilhelm May

Centre for Environmental and Climate Research, Lund University, Lund, SE-223 62, Sweden

*Correspondence to*: Wilhelm May (Wilhelm.may@cec.lu.se)

**Abstract.**

In this study, the role that more realistic soil moisture has for the characteristics of surface energy fluxes in two sets of reanalyses performed at ECMWF is investigated. These are the "standard" set of reanalyses ERA-Interim ("ERAInt") and the ERA-Interim/Land reanalyses of the land surface conditions ("ERAInt/Land"). In the latter, the ECMWF's land surface model

has been forced with the meteorological fields from ERAInt, including an adjustment of precipitation based on the monthly mean values from the Global Precipitation Climatology Project data set. Adjusting precipitation has a distinct impact on the soil moisture content in the two sets of reanalyses. ERAInt is characterized by a general tendency to underestimate (overestimate) soil moisture in regions with a relatively high (low) soil moisture content. The differences in soil moisture between ERAInt and ERAInt/Land vary only slightly in the course of the year. This is not the case for precipitation, where the

differences between the two sets of reanalyses vary markedly between different seasons. The direct impact of the regional differences in precipitation between ERAInt and ERAInt/Land on the corresponding deviations in soil moisture varies considerably by region. One reason is that the regional differences in precipitation vary by season, while the regional differences in soil moisture typically persist throughout the year. Another reason is that the specific nature of the interaction between precipitation and soil moisture diverges between different regions, depending on the climate conditions and on the

degree to which the soil is saturated with moisture. The differences in soil moisture between the two sets of reanalyses have notable effects on the characteristics of surface energy fluxes. The nature of these effects differs by region and also by season, that is the coupling between soil moisture and the latent or the sensible heat flux is positive in one region or season, respectively, and negative in another one. In any case, the differences in the soil moisture content typically affect the latent and the sensible heat flux in opposite ways. Increases (decreases) in latent heat flux typically coincide with decreases (increases) in sensible

heat flux. By this, the differences in soil moisture have a substantial impact on the partitioning of latent and sensible heat flux. The effect of the soil moisture differences on the evaporative fraction, for instance, is mainly governed by the impact on the latent heat flux because of the opposite effects on latent and sensible heat fluxes and, hence, only a weak impact on the total surface energy flux. The effect on the Bowen ratio, on the other hand, is for the most part controlled by the impact on the sensible heat flux, with higher (lower) values of the Bowen ratio in regions with increased (decreased) sensible heat flux. The

representation of precipitation in ERAInt is governed by the characteristics of ECMWF's Integrated Forecasting System ("IFS"). As the EC-Earth coupled climate model incorporates the IFS atmospheric general circulation model, the comparison



between ERAInt and ERAInt/Land serves as a reference point for the simulation of the soil moisture content and of surface energy fluxes by EC-Earth.

## 1 Introduction

The coupling The coupling between the land surface and the atmosphere has a profound impact on the water and energy cycles,
in particular through the feedbacks between soil moisture and precipitation and between soil moisture and temperature. Through its impact on the separation of latent and sensible heat flux, soil moisture affects the state of the atmosphere (air temperature, boundary layer stability or precipitation) and, thus the characteristics of regional weather and climate (e.g., Seneviratne et al., 2010).

As for the coupling with temperature, a negative soil moisture anomaly causes decreasing evapotranspiration. This
leads to an increased sensible heat flux and, hence, to warmer near-surface air temperatures. Warmer temperatures, in turn, cause a further decrease in soil moisture due to a potential increase in evapotranspiration, in response to a higher water vapour deficit and an enhanced evaporative demand. A positive soil moisture anomaly, on the other hand, leads to increasing evapotranspiration and a decreasing sensible heat flux, resulting in colder air temperatures. The near-surface cooling, in turn, stabilizes the atmospheric boundary layer and therefore, might, suppress convective precipitation. The latter counteracts the
initial positive soil moisture anomaly. As for the coupling with precipitation, a negative (positive) soil moisture anomaly generally leads to less (more) evapotranspiration and, thus, a decreased (increased) latent heat flux. The impact of the induced changes in latent heat flux on precipitation is somewhat uncertain, as several complex processes are involved. This uncertainty is related to both the direction of the coupling, is the coupling positive with a negative or a positive anomaly in evapotranspiration leading to decreased (increased) precipitation or negative with an increase (decrease) in precipitation in
response to a deficit (excess) in evapotranspiration, and its strength. Increasing (decreasing) precipitation typically causes more (less) soil moisture, with some exceptions. In the case of precipitation over wet or saturated soils, for instance, precipitation anomalies do not result in anomalous soil moisture but anomalous runoff. In the case of a strong drought, on the other hand, precipitation bypasses the surface zone through infiltration.

The relationship between soil moisture and climate has been particularly addressed in the Global Land Atmosphere
Coupling Experiment ("GLACE"; e.g., Koster et al. 2004; 2010). In  GLACE and other similar experiments the land surface states were generally obtained in offline simulations with the respective land surface models, which are forced with meteorological fields from atmospheric reanalyses. In some cases, simple schemes correcting for potential climatic biases are employed. It is particularly important to correct for biases in precipitation in order to ensure consistency for the land hydrology. In recent years, several improved global atmospheric reanalyses have become available. These are, for instance NASA's
Modern-Era Retrospective Analysis for Research and Applications ("MERRA"; Rienecker et al., 2011) and the European Centre for Medium-Range Weather Forecasts ("ECMWF") Interim reanalyses ("ERA-Interim"; Dee et al., 2011). Combined



with observational estimates of precipitation, these two reanalyses have been used to derive improved land surface products, i.e., "MERRA-land" (Reichle et al.; 2011) and "ERA-Interim/Land" (Balsomo et al; 2015), respectively. ERA-Interim/Land is the result of a stand-alone simulation with the 2013-version of the ECMWF land surface model. The land surface model has been forced with the meteorological fields from ERA-Interim, but adjusting precipitation using monthly mean estimates of

observed precipitation. A comparison with available observations revealed that the state of the land surface in ERA-Interim/Land is considerably improved as compared to the original ERA-Interim reanalysis (Balsamo et al., 2015). The authors concluded that ERA-Interim/Land provides globally consistent estimates of soil moisture (as well as snow water equivalent), suitable for initialising numerical weather prediction or climate models.

The reanalyses undertaken at ECMWF incorporate the Integrated Forecasting System ("IFS") atmospheric general
circulation model ("AGCM"; e.g., Tiedtke, 1989; 1993; Hortal et al., 2002; Bechtold et al.; 2008, Morcrette et al., 2008). IFS is also the atmospheric component of ECMWF's seasonal forecast system (Molteni et al., 2011) and of the EC-Earth coupled atmosphere-ocean-sea-ice climate model (Hazeleger et al., 2012), which is suitable for extended climate simulations. IFS has some specific errors in the simulation of precipitation, as many other numerical climate models, which typically show modest agreement with observations with regard to broad-scale features but important systematic errors in the tropics (Flato et al.,
2013). Errors in the simulation of precipitation affect the simulation of soil moisture, with a positive (negative) precipitation bias generally resulting in a too high (low) soil moisture content. Such soil moisture anomalies alter evaporation (the latent heat flux) as well as the partitioning of latent and sensible heat flux at the land surface. These changes in the surface energy fluxes do not only affect the state of the atmosphere, but might also cause precipitation anomalies. In the case that positive (negative) soil moisture anomalies lead to more (less) precipitation, the coupling between soil moisture and precipitation
maintains the initial errors in the simulation of precipitation in a numerical climate model.

In this study, the two kinds of reanalyses provided by ECMWF are used to study the role that a more realistic soil moisture has for the characteristics of surface energy fluxes in two sets of reanalyses provided by ECMWF. In ERA-Interim/Land precipitation has been adjusted by observational data, considerably improving the representation of the soil moisture content compared to the standard set of reanalyses, ERA-Interim. In order to assess the impact of the improved soil
moisture on the characteristics of surface energy fluxes in the reanalyses, the differences in the latent and sensible heat flux between ERA-Interim and ERA-Interim/Land are related to the corresponding deviations in the soil moisture content. Furthermore, the role of the more realistic soil moisture for the partitioning between latent and sensible heat fluxes is investigated. In addition to geographical distributions, four selected regions are studied in further detail, i.e., the coastal and the Sahel region in West Africa, India (combined with Bangladesh and Sri Lanka) and the Amazonas catchment.

The two sets of reanalyses considered in this study are described in Section 2. In Section 3, the impact of adjusting precipitation using observational data on the reanalysis of soil moisture in investigated. In Section 4, the role of the resulting



differences in the soil moisture conditions for the surface energy fluxes and the partitioning between latent and sensible heat fluxes is analysed. A summary and conclusions are given in Section 5.

## 2 Data

This study is based on two kinds of reanalyses provided by ECMWF over the period 1979-2010, namely ERA-Interim ("ERAInt"; Dee et al., 2011) and ERA-Interim/Land ("ERAInt/Land"; Balsamo et al., 2015). ERAInt/Land is a global land surface reanalysis data set, which describes the evolution of soil moisture, soil temperature and the snowpack. It is the result of a stand-alone simulation with the 2013-version of the ECMWF land surface model ("HTESSEL"; Balsamo et al., 2009), driven by meteorological forcing from ERAInt, including a precipitation adjustment based on monthly mean values from version 2.1 of the Global Precipitation Climatology Project data set ("GPCP"; Huffman et al., 2009). The time step of the simulation is 3 hours, which is the temporal resolution of ERAInt. Compared to the previous version of the ECMWF land surface model ("TESSEL"; van den Hurk et al., 2000), which has been used for ERAInt, HTESSEL includes several changes. These are an improved soil hydrology, a new snow scheme (Dutra et al., 2010), a multiyear satellite-based vegetation climatology (Boussetta et al., 2013) and a revised bare soil evaporation (Albergel at al., 2012). In the Northern Hemisphere, the improvement in ERAInt/Land as compared to ERAInt could be attributed to the revisions of land parameterizations (Balsamo et al., 2015). In the tropics and in the Southern Hemisphere, on the other hand, the correction of precipitation was found to be important. In both versions of the ECMWF land surface model, soil moisture is analysed for four soil layers, i.e., between 0 and 7 cm, between 7 and 28 cm, between 28 and 100 cm and between 100 and 255 cm depth below land. Here, the soil moisture content integrated over the uppermost meter (the three uppermost layers), representing the root-zone, is considered.

Here, reanalyses of the daily values of soil moisture from both ERAInt and ERAInt/Land are used. In addition to the soil moisture content, daily estimates of accumulated precipitation and of the latent and sensible heat flux are considered. For ERAInt/Land, these estimates originate from the continuous stand-alone simulation, where HTESSEL is forced with the meteorological fields from ERAInt (including a precipitation adjustment) as described above. For ERAInt, on the other hand, the daily estimates of precipitation and the surface energy fluxes originate from short-term forecasts initiated from the reanalyses of the prognostic meteorological variables twice a day (00 and 12 UTC; Dee et al., 2011). Here, the 24-hour forecasts of accumulated precipitation and of the latent and sensible heat flux initiated at 00 UTC are used. This means that the complete daily cycle is included, regardless of the location on the globe. All the data have a horizontal resolution of about 80 km with 256×512 grid points distributed over the globe.



## 3 Soil moisture and precipitation

In this section, the impact of adjusting precipitation using observational data on the reanalyses of soil moisture is investigated. This is done by comparing the soil moisture from the two sets of reanalyses and relating differences in the soil moisture content to the deviations in mean precipitation for the two sets of reanalyses.

### 3.1 Geographical distributions

Geographical distributions of long-term seasonal means of soil moisture and precipitation over the period 1979-2010 are presented for two seasons, namely boreal summer including June through August ("JJA") and austral summer containing December through February ("DJF"). These distributions cover the area between 60° N and 60° S in order to exclude those parts of the globe that are covered with snow or ice throughout a large part of the year.

Figures 1a, b show the long-term seasonal means of the soil moisture content in the uppermost meter for ERAInt/Land in the two seasons. The geographical distributions reveal that in both seasons the soil moisture content is rather high in the tropics (including South and Southeast Asia), the Northern Hemisphere mid-latitudes (except for the south-western part of North America) and over the La Plata basin in South America. Relatively small values of soil moisture are found in the subtropics, Central Asia and the south-western part of North America. Despite the overall similarity of the geographical distributions for the two seasons, some seasonal variations are found (Fig. 1e). In JJA, the soil moisture content is particularly high in the northern part of the tropics as well as in South and Southeast Asia, also over the southern parts of South America and Australia. In DJF, on the other hand, the values of the soil moisture are relatively high in the southern part of the tropics as well as in North America, Europe and western Russia.

As indicated by the differences between ERAInt and ERAInt/Land, ERAInt has a tendency to underestimate soil moisture in the tropics, including South and Southeast Asia, over the La Plata basin and over the eastern part of Australia (Figs. 1c, d). Also over the south-eastern part of North America, in Europe (only in DJF, though) and in northern Asia, the soil moisture content is too low. Soil moisture is overestimated in northern Africa, the Middle East and Central Asia as well as over the south-western part of North America, the south-western part of Africa and the western and central parts of Australia. That is, ERAInt has a general tendency to underestimate (overestimate) soil moisture in areas with a relatively high (low) soil moisture content. The geographical distributions of the differences between the two sets of reanalyses diverge to some extent between the two seasons (Fig. 1f). In DJF, for instance, the underestimation of soil moisture is particularly strong over the southern part of Amazonia, most of Southern Africa, the south-eastern part of North America and western Europe. In JJA, on the other hand, the underestimation is more pronounced over the very northern part of South America, the coastal region of West Africa and equatorial Africa as well as in South and Southeast Asia. The comparison with the corresponding seasonal differences for ERAInt/Land (see Fig. 1e) reveals that the seasonal variation of the soil moisture content is noticeably weaker in ERAInt than in ERAInt/Land (indicated by corresponding differences of the opposite sign in the two panels).



The seasonal variation of the geographical distributions of soil moisture is largely accounted for by corresponding seasonal differences in mean precipitation. In JJA, precipitation is rather strong around the equator and in the northern part of the tropics as well as in South and Southeast Asia (Fig. 2a). In DJF, on the other hand, precipitation is particularly strong near the equator and in the southern part of the tropics as well as over the maritime continent (covering the southern part of Malaysia,

Indonesia, the Philippines and New Guinea; Fig. 2b). These noticeable seasonal variations in precipitation can be found again in the geographical distribution of the difference in soil moisture between JJA and DJF (see Fig. 1e), supporting the finding that seasonal variations in precipitation are the main cause of the seasonal differences in soil moisture at a regional scale.

The geographical distributions reveal some marked differences in precipitation between the two sets of reanalyses (Figs 2c, d). In South America, for instance, ERAInt is characterized by more precipitation than ERAInt/Land over the southern part

of Amazonia and less precipitation over the northern part of the continent in JJA (Fig. 2c). In DJF, on the other hand, ERAInt shows less precipitation over most of the tropical and subtropical parts of South America (Fig. 2d). This means that in large parts of South America ERAInt overestimates (underestimates) precipitation in regions with weak (strong) precipitation in the respective season. In both seasons, ERAInt is characterized by more precipitation on the coast of West Africa and over equatorial Africa as well as over parts of Southeast Asia. In JJA, ERAInt underestimates precipitation in West Africa except

near the coast as well as in India (Fig. 2c). In DJF, on the other hand, ERAInt underestimates precipitation over the south-eastern part of North America, in Europe and over western Russia (Fig. 2d). In both seasons, ERAInt is characterised by less precipitation over the maritime continent. The geographical distributions of the differences between the two sets of reanalyses have opposite signs for the two seasons over many parts of the global land areas. Notable exceptions are the overestimations of precipitation in ERAInt in equatorial Africa, over parts of Southeast Asia and over the southern part of North America in

both seasons (Figs. 2c, d). At the same time, ERAInt underestimates precipitation in both seasons over the western part of Europe, the southern part of Australia and over the maritime continent in both seasons.

Figure 2e illustrates the aforementioned seasonal variations of the differences between the two sets of reanalyses. The comparison with the corresponding figure for the soil moisture content (see Fig. 1f) shows that the geographical distributions are very similar, with corresponding negative or positive deviations of the two quantities in many regions. This indicates that

the seasonal variations of the differences in soil moisture between ERAInt and ERAInt/Land in these regions are mainly caused by corresponding seasonal variations of the differences in precipitation. The only marked exception is the southern part of Africa with positive (negative) deviations for soil moisture (precipitation). The negative differences of the soil moisture content are due to a relatively weak underestimation of soil moisture in ERAInt in DJF (see Fig. 1d), while the positive differences in precipitation are related to a strong overestimation of precipitation in ERAInt in this season (Fig. 2d).

Despite the importance of precipitation for the seasonal variation of the differences in the soil moisture content, the situation is different for the differences between the two sets of reanalyses for the individual seasons. This is illustrated by the notable disparities of the geographical distributions presented in Figures 1c and 2c and in Figures 1d and 2d, respectively. One



reason for this is that for the soil moisture content the differences between ERAInt and ERAInt/Land have very similar geographical structures in the two seasons, while for precipitation the corresponding distributions vary considerably between the two seasons. The other reason is that the effect of the adjusted precipitation on soil moisture is governed by the specific nature of the interaction between precipitation and soil moisture.

## 3.2 Regional perspective

In addition to the geographical distributions, area averages are shown for four selected regions covering the entire annual cycle via 12 seasons. These seasons are defined as overlapping 3-month periods ranging from January through March ("JFM"), over February through April ("FMA"), etc., to DJF. These 12 seasons (rather than, for instance, the four climatological seasons) are chosen in order to obtain a smoother representation of the annual cycle. These regions (only land areas are included) are two areas in West Africa, the coastal region ("WAF-C") and the Sahel region ("WAF-S"). The other areas are the Indian region and the Amazonas catchment ("AMZ"; e.g., Boisier et al. 2015). See Table 1 for the longitude and latitude ranges of the four regions.

These four areas are characterized by pronounced variations of the accumulated seasonal precipitation in the course of the year in accordance with the meridional shift of the area with convective activity. In WAF-S (Fig. 3a), WAF-C (Fig. 3c) and IND (Fig. 3e), which are located in the Northern Hemisphere, the amount of precipitation is particularly high during boreal summer, while in AMZ (Fig. 3f), which is mainly located in the Southern Hemisphere, precipitation is relatively strong during boreal summer. The soil moisture content shows notably weaker variations in the course of the year, with rather high (low) values of soil moisture typically occurring slightly (about one month) later than the season with rather strong (weak) precipitation.

In the Sahel (Fig. 3b) and in the Indian region (Fig. 3f), ERAInt notably underestimates precipitation during boreal summer but gives a realistic amount of precipitation during winter, indicating a weaker annual cycle of precipitation in these areas. Consistent with this, the soil moisture content is underestimated in IND during the second half of the year, with the strongest underestimation at the end of the Indian summer monsoon season. In WAF-S, on the other hand, ERAInt overestimates soil moisture in all seasons, with a relatively weak underestimation at the end of the West African monsoon season, though. In the coastal region of West Africa, ERAInt overestimates precipitation but underestimates the soil moisture content throughout the year (Fig. 3d). Also in the Amazonas catchment, soil moisture is underestimated in ERAInt throughout the year, while precipitation is overestimated during boreal summer and underestimated in austral summer (Fig. 3h). Similar to WAF-S and IND, the annual cycle of precipitation is weaker in ERAInt in AMZ. The annual cycles of soil moisture, on the other hand, are underestimated in ERAInt in all four areas.



## 4 Surface energy fluxes

Soil moisture plays not only an important role for the water cycle but also for the energy cycle through its impact on the partitioning of the energy fluxes at the land surface. Therefore, in this section, the impact of the adjusted precipitation on the characteristics of the surface energy fluxes is investigated, including the effect on the separation of latent and sensible heat
fluxes.

### 4.1 Latent and sensible heat flux

The geographical distributions of the latent heat flux are characterized by distinct seasonal variations. In JJA, latent heat fluxes are rather strong around the equator and in the northern part of the tropics as well as in South and Southeast Asia (Fig. 4a). Latent heat fluxes are also relatively strong in the eastern U.S. and Europe, extending into Russia. In DJF, on the other hand,
latent heat fluxes are particularly strong near the equator and in the southern part of the tropics as well as over the south-eastern part of South America and the eastern part of Southern Africa (Fig. 4b).

In JJA, ERAInt gives higher values of the latent flux than ERAInt/Land over most areas (Fig. 4c). Marked positive differences are found in Amazonia, Central Africa and in North America, over parts of Europe and Asia as well as in Southern Africa. Distinct negative differences, indicating weaker latent heat fluxes in ERAInt, are located in West Africa and over the
western part of India. In DJF, on the other hand, ERAInt gives higher values of the latent heat flux mainly in parts of South America, sub-Saharan Africa as well as in South and Southeast Asia (Fig. 4d). In most of the Northern Hemisphere extratropics latent heat fluxes are slightly reduced in ERAInt.

Also the geographical distributions of the sensible heat fluxes show marked seasonal variations. In JJA, sensible heat fluxes are rather strong over the western part of North America, in northern Africa and Central Asia as well as over the
subtropical part of Southern Africa (Fig. 5a). In DJF, on the other hand, sensible heat fluxes are relatively strong over the very southern parts of South America and Africa as well as in most of Australia except for the very north (Fig. 5b). Over most of the Northern Hemisphere mid-latitudes, the values of the sensible heat flux are negative in DJF (see also Fig. 7b), meaning that the land surface gains sensible heat from the atmosphere.

In contrast to the latent heat fluxes, ERAInt is characterised by lower values of the sensible heat flux than ERAInt/Land
over most areas in JJA (Fig. 5c). Notable negative deviations are found over North America, in Central Asia as well as over the Amazonas catchment and in equatorial Africa. Higher values of the latent heat flux occur mainly in northern Africa, over the southern part of India and over the south-eastern part of South America. In DJF, ERAInt gives weaker sensible heat fluxes over some parts of the Northern Hemisphere mid-latitudes and, particularly, over the south-western part of South America, Africa south of about 10 °N and the south-eastern part of Australia (Fig. 5d). Higher values of the sensible heat flux are found
over other parts of the Northern Hemisphere mid-latitudes, in northern Africa and over the Arabian Peninsula as well as over the north-western part of Australia.



In order to analyse, to which extent the differences in the surface energy fluxes are related to the differences in soil moisture between ERAInt and ERAInt/Land, non-centred correlations between the time series of the seasonal mean differences in the latent and sensible heat fluxes, respectively, and in the soil moisture content over the period 1979-2010 have been computed.

5     Figure 6 illustrates not only the extent to which anomalous soil moisture contributes to the differences in the surface energy fluxes but also the sign of the coupling between the respective meteorological variables. In JJA, notable negative correlations between latent heat flux and soil moisture are found in Amazonia, the Sahel region (in combination with negative correlations on the Guinea Coast), Central Africa and Southeast Asia (Fig. 6a). Notable positive correlations are confined to the extratropics, i.e., the southern parts of South America and Southern Africa as well as Europe and western Russia. In DJF, 10     negative correlations are mainly found in the tropics, i.e., in Central Africa and Southeast Asia as well as over the maritime continent (Fig. 6b). Similar to JJA, also in DJF positive correlations occur mainly in the extratropics. For the sensible heat flux (Figs. 6c, d) the correlations with soil moisture generally have the opposite sign and a similar magnitude as for the latent heat flux. This illustrates the overall tendency that soil moisture anomalies affect the sensible heat fluxes in opposite ways, leading to higher (lower) values of sensible heat fluxes in regions, where the latent heat flux is reduced (increased).

15 **4.2 Partitioning of surface energy fluxes**

The partitioning between the surface energy fluxes is described using two different measures. The Bowen ratio ("BR") is defined as the ratio between the sensible and the latent heat flux. For values less than one, a larger fraction of the energy available at the land surface is passed to the atmosphere as latent heat, for values larger than one the opposite is true. In the case of very low values of the latent heat flux, such as in arid regions, BR becomes unbound and another measure of the relative 20     contributions of the surface energy fluxes to the energy budget at the land surface is more appropriate. This is the evaporative fraction ("EF"), which is defined as the ratio between the latent energy flux and the total surface energy flux (the sum of the latent and the sensible heat flux). This means that EF and BR are related through the equation EF=1/(1+BR).

Regions with high values of BR (exceeding 5) are located in the subtropics (Figs. 7a, b). In JJA, these regions are the south-western part of North America, northern Africa, extending over the Arabian Peninsula, Pakistan and Afghanistan to the 25     Tibetan Plateau, as well as Southern Africa and the northern part of Australia (Fig. 7a). Particularly low values (below 0.5) occur in the tropics and in the Northern Hemisphere extratropics. In DJF, the regions with high values of BR in the Northern Hemisphere are smaller than in JJA, and in the Southern Hemisphere, such high values mainly occur over the southern part of Australia (Fig. 7b). In the mid-latitudes of the respective winter hemisphere, BR is negative, as the sensible heat fluxes are directed from the atmosphere to the land surface.

30     The differences in the soil moisture content between the two sets of reanalyses have a notable impact on BR (Figs. 7c, d). In JJA, ERAInt gives higher values of BR over the Sahara, the Arabian Peninsula and the Tibetan Plateau (Fig. 7c). Over





much of the rest of the Northern Hemisphere, BR is weaker in ERAInt, particularly over the south-western part of North America and in Pakistan and Bangladesh. Also in the Southern Hemisphere, ERAInt is characterized by lower values of BR in all regions except for the south-western part of Australia. The reduction is especially strong in Southern Africa. Also in DJF, BR is considerably enhanced over the Sahara and the Arabian Peninsula (Fig. 7d). BR is also enhanced over the western part

of Australia but reduced to the east, and it is slightly reduced over Africa south of about 20 °N.

Because of the aforementioned relation between the two measures, the geographical distributions of EF (Figs. 8a, b) are complementary to the distributions of BR, with high (low) values of EF typically occurring in regions with low (high) values of BR. Particularly high (low) values of EF are found in those regions, where the soil moisture content is rather high (low; see Figs. 1a, b), which, in turn. lead to strong (weak) fluxes of latent heat (see Figs. 4a, b). Therefore, also the effect of

the adjusted precipitation on EF (Figs. 8c, d) is governed by its effect on the latent heat flux (see Figs. 4c, d), with negative or positive differences between ERAInt and ERAInt/Land in the same regions. This means that the total energy flux is much less affected by the adjusted precipitation as the latent or the sensible heat flux, consistent with the reversed impacts on latent and sensible heat fluxes (see above).

## 4.3 Regional perspective

In the four regions considered here, the values of BR vary substantially in the course of the year. In the Sahel region (Fig. 9a) and the Indian region (Fig. 9e), the values of BR are rather low in boreal summer but much higher during winter. These variations mainly reflect pronounced annual cycles of the latent flux with rather strong fluxes in boreal summer and weaker fluxes during winter, combined with relatively weak sensible heat fluxes in boreal summer, particularly in IND. In the coastal region of West Africa, on the other hand, it is mainly the seasonal variation of the sensible heat flux, which determines the

seasonal variation of BR with relatively low (high) values in boreal summer (winter; Fig. 9c). Also in the Amazonas catchment, it is the seasonal variation of the sensible heat flux, which governs the seasonal variation of BR with relatively low values during austral summer and higher values in winter (Fig. 9g).

In WAF-S, BR is significantly stronger in ERAInt than in ERAInt/Land during boreal summer and weaker in winter (Fig. 9b). This is related to stronger sensible but weaker latent heat fluxes in summer and the reverse behaviour in winter. In

IND the situation is similar, but the values of BR are only slightly higher in ERAInt during boreal summer, because the sensible heat flux is only slightly enhanced during this part of the year and the latent heat flux hardly differs between the two sets of reanalyses (Fig. 9f). In WAF-C, BR is weaker in ERAInt throughout the entire year, but more in boreal winter than is summer (Fig. 9d). The overall tendency is related to the fact that the positive differences in the latent heat flux exceed the corresponding negative differences in the sensible heat flux. The seasonal variation of the difference in BR, on the other hand, is mainly

associated with a corresponding seasonal variation of the differences in the sensible heat flux. In AMZ, the negative differences of BR during austral winter indicate that the seasonal variation of BR is weaker in ERAInt than for ERAInt/Land (Fig. 9h).



This is mainly due to weaker sensible heat fluxes and stronger latent heat fluxes over the Amazon catchment during the respective part of the year.

The annual cycles of EF in the four regions follow essentially the corresponding annual cycles of the latent heat flux (Figs. 10a, c, e, g). There are some deviations in the amplitudes of the respective annual cycles in relation to the seasonal variation of the sensible heat flux. Also the differences of EF between the two sets of reanalyses in the four regions are in accordance with the corresponding differences in the latent heat flux (Figs. 10b, d, f, h). This reflects the general tendency that the differences for the latent and the sensible heat flux have opposite tendencies, so that the differences are rather small for the total energy fluxes. The Indian region, though, is an exception. During boreal summer, the positive difference in the sensible heat flux exceeds the corresponding difference in the latent heat flux, resulting in smaller values of EF in ERAInt (Fig. 10f).

## 5 Summary and conclusions

In this study, the role that more realistic soil moisture has for the characteristics of surface energy fluxes in two sets of reanalyses performed at ECMWF is investigated. These are the "standard" set of reanalyses ERA-Interim ("ERAInt"; e.g., Dee et al., 2011) and the ERA-Interim/Land reanalyses of the land surface conditions ("ERAInt/Land"; Balsamo et al., 2015). In this stand-alone simulation, the ECMWF's land surface model has been forced with the meteorological fields from ERAInt, including an adjustment of precipitation based on the monthly mean values from version 2.1 of the Global Precipitation Climatology Project data set (Huffman et al., 2009).

Adjusting precipitation has a distinct impact on the soil moisture content in the two sets of reanalyses. ERAInt reveals a general tendency to underestimate (overestimate) soil moisture in regions with a relatively high (low) soil moisture content. ERAInt is, for instance, characterized by too high values of soil moisture in the tropics and parts of the extratropics and too low values in the subtropics. These differences in soil moisture between ERAInt and ERAInt/Land vary only slightly in the course of the year. This is not the case for precipitation, where the differences between the two sets of reanalyses vary markedly between different seasons. During boreal summer, for instance, ERAInt underestimates precipitation in Central America and over the northern part of Amazonia as well as in the Sahel region and over the Indian subcontinent, but overestimates precipitation over the southern part of Amazonia as well as on the Guinea Coast and in Equatorial Africa. During austral summer, on the other hand, precipitation is underestimated over entire Amazonia and, similar to boreal summer, overestimated on the Guinea Coast and in Equatorial Africa. The direct impact of the regional differences in precipitation between ERAInt and ERAInt/Land on the corresponding deviations in soil moisture varies considerably by region. One reason is that the regional differences in precipitation vary by season, while the regional differences in soil moisture typically persist throughout the year. Another reason is that the specific nature of the interaction between precipitation and soil moisture diverges between different regions, depending on the climate conditions and on the degree to which the soil is saturated with moisture.



The differences in the soil moisture content between the two sets of reanalyses have notable effects on the characteristics of surface energy fluxes. The nature of these effects differs by region and also by season, that is the coupling between soil moisture and the latent or the sensible heat flux is positive in one region or season, respectively, and negative in another one. In boreal summer, for instance, the overestimated soil moisture in Amazonia and in Equatorial Africa leads to stronger latent

heat fluxes in these regions, while in Southern Africa the underestimated soil moisture results in weaker latent heat fluxes. In the Sahel region, on the other hand, the overestimated soil moisture leads to weaker latent heat fluxes. In boreal winter, on the other hand, the overestimated soil moisture over the southern part of the Sahel region leads to stronger latent heat fluxes. The differences in the soil moisture content typically affect the latent and the sensible heat flux in opposite ways, i.e., increases (decreases) in latent heat flux coincide with decreases (increases) in sensible heat flux. By this, the differences in soil moisture

have a substantial impact on the partitioning of latent and sensible heat flux. The effect of the soil moisture differences on the evaporative fraction (the ratio between the latent heat flux and the total energy flux) is mainly governed by the impact on the latent heat flux because of the opposite effects on latent and sensible heat fluxes and, hence, only a weak impact on the total surface energy flux. The effect on the Bowen ratio (the ratio between the sensible and the latent heat flux), on the other hand, is for the most part controlled by the impact on the sensible heat flux, with higher (lower) values of the Bowen ratio in regions

with increased (decreased) sensible heat flux.

As ERAInt/Land is forced with the meteorological data from ERAInt, the differences in the characteristics of the surface energy fluxes between the two sets of reanalyses are caused by the differences in the characteristics of the land surface conditions in relation to the adjusted precipitation, the adjusted snow cover or some updates of the ECMWF land surface model. As for the improved soil moisture content, the effect of the adjusted precipitation is particularly important. Therefore,

also the differences in the latent heat flux and, thus, the differences in the sensible heat flux are mainly related to the adjusted precipitation. The differences in precipitation, in turn, depend on the short-term predictions of precipitation in ERAInt and, therefore, on this particular aspect of ECMWF's IFS. As the EC-Earth coupled climate model incorporates the IFS AGCM, the simulation of precipitation by EC-Earth will inherit the initial deficiencies in the representation of precipitation in IFS. In contrast to ERAInt, in simulations with EC-Earth the problems with the representation of precipitation will not only affect the

simulation of soil moisture. Because of the two-way coupling between the atmosphere and the land surface, the induced soil moisture anomalies will feed back on the state of boundary layer, the atmospheric circulation and on the simulations of precipitation itself. As a result, the representation of precipitation in EC-Earth might be less realistic than in ERAInt. Therefore, the comparison between ERAInt and ERAInt/Land serves as a reference point for the simulation of the soil moisture content and of surface energy fluxes by EC-Earth.

The results presented here show that differences in the simulated soil moisture content have a notable impact on the surface energy fluxes as well as on the partitioning of latent and sensible heat fluxes. By this, deviations in soil moisture not only affect near-surface temperatures, the stability of the boundary layer, clouds and precipitation locally. They may also have remote effects by altering the sea-level pressure and, by this, the large-scale circulation in a particular region. This emphasizes



the importance of a realistic simulation of soil moisture and, more general, a comprehensive and realistic representation of land surface processes in numerical climate models. This is not only a requirement for more realistic climate simulations but also a prerequisite for more reliable future climate scenarios (van den Hurk et al., 2016).

**Acknowledgements**

The research presented in this paper is a contribution to the Swedish strategic research area Modelling the Regional and Global Earth System, MERGE. Thanks to the European Centre for Medium-Range Weather Forecasts, ECMWF, for the provision of the ERA-Interim and ERA-Interim/Land re-analysis data. Access to the computer facilities at ECMWF is granted through the special project on "Coupling and feedbacks between soil moisture and two dominant monsoon systems".

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





**Tables**

| Acronym | Region | Longitude range | Latitude range |
|---|---|---|---|
| **WAF-S** | West Africa – Sahel region | 10 °W – 10 °E | 10 °N – 20 °N |
| **WAF-C** | West Africa – coastal region | 10 °W – 10 °E | 10 °N – 20 °N |
| **IND** | Indian region incl. India (except for the mountainous northern part), Bangladesh and Sri Lanka | 70 °E – 94 °E | 6 °N – 32 °N |
| **AMZ** | Amazonas catchment | 79 °W – 45 °W | 20 °S – 5 °N |

**Table 1: Definition on the four regions presented in this study.**

**Figures**





**Figure 1: Seasonal mean soil moisture content in the uppermost meter for ERAInt/Land in (a) boreal summer (JJA; June through August) and (b) austral summer (DJF; December through February). Also the differences in the seasonal mean soil moisture content between ERAInt and ERAInt/Land in (c) JJA and (d) DJF as well as the differences between the seasonal means in JJA and DJF for (e) ERAInt and (f) the difference between ERAInt and ERAInt/Land. Units are cm; the contour interval is 2 cm.**



**Figure 2:** Seasonal mean precipitation for ERAInt/Land in (a) boreal summer (JJA) and (b) austral summer (DJF). Also the differences in the seasonal mean precipitation between ERAInt and ERAInt/Land in (c) JJA and (d) DJF as well as (e) the differences between the seasonal means in JJA and DJF for the difference between ERAInt and ERAInt/Land. Units are mm/day; the contour intervals are 0.5 mm/day (a, b) and 0.25 mm/day (c-e), respectively.











**Figure 3: Seasonal mean soil moisture content in the uppermost meter and the accumulated sesaonal precipitation averaged over four regions in 12 seasons (see Table 1) for ERAInt/Land (left column) and the corresponding differences between ERAInt and ERAInt/Land (right column). Units are cm.**

30




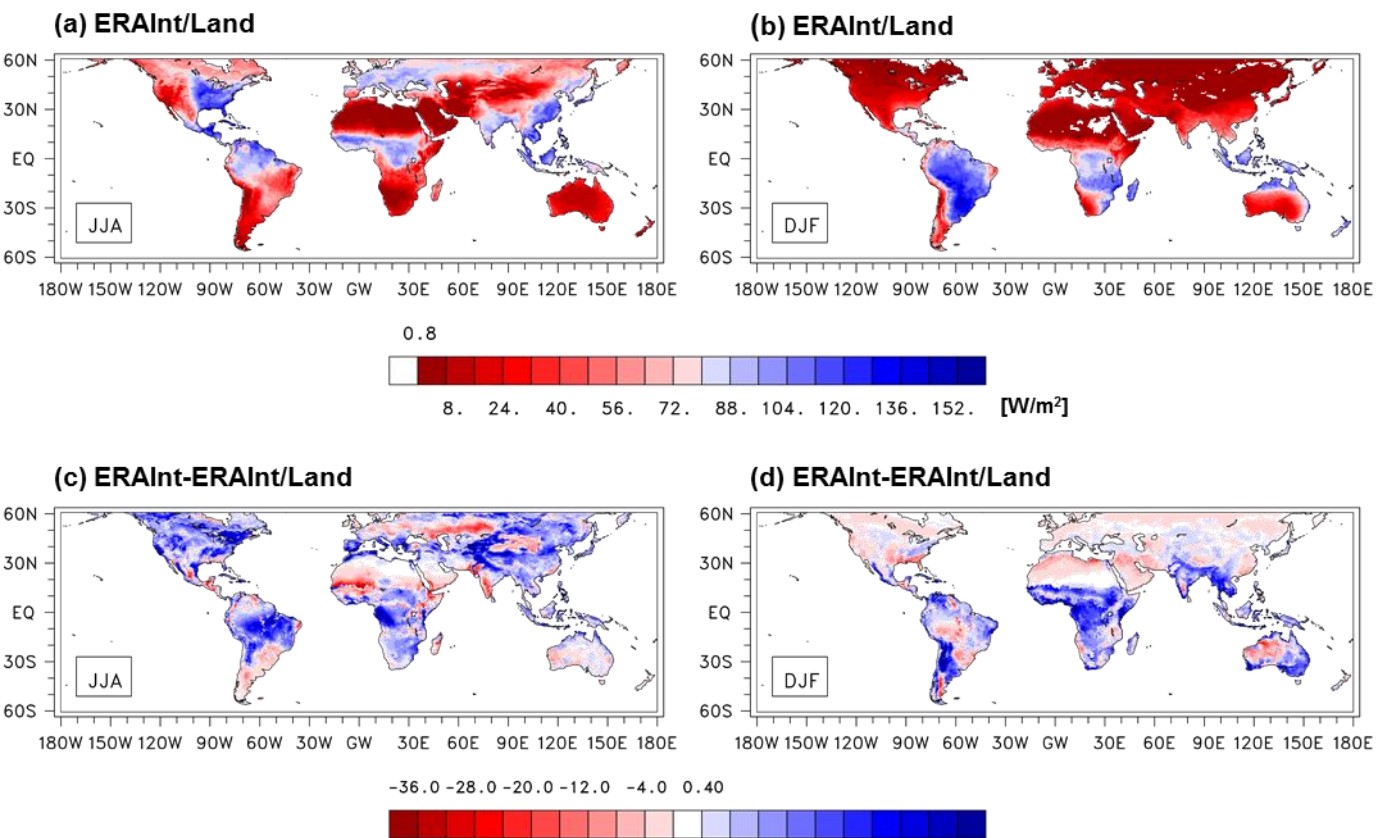

**Figure 4: Seasonal mean latent heat flux for ERAInt/Land in (a) boreal summer (JJA) and (b) austral summer (DJF). Also the differences in the seasonal mean latent heat flux between ERAInt and ERAInt/Land in (c) JJA and (d) DJF. Units are W/m²; the contour intervals are 8 W/m² (a, b) and 4 W/m² (c, d), respectively. Fluxes into the atmosphere are positive.**




**Sensible heat flux**

**(a) ERAInt/Land**   **(b) ERAInt/Land**

**(c) ERAInt-ERAInt/Land**   **(d) ERAInt-ERAInt/Land**

Figure 5: As Fig. 4 but for the sensible heat flux. Units are W/m²; the contour intervals are 8 W/m² (a, b) and 4 W/m² (c, d), respectively. Fluxes into the atmosphere are positive.





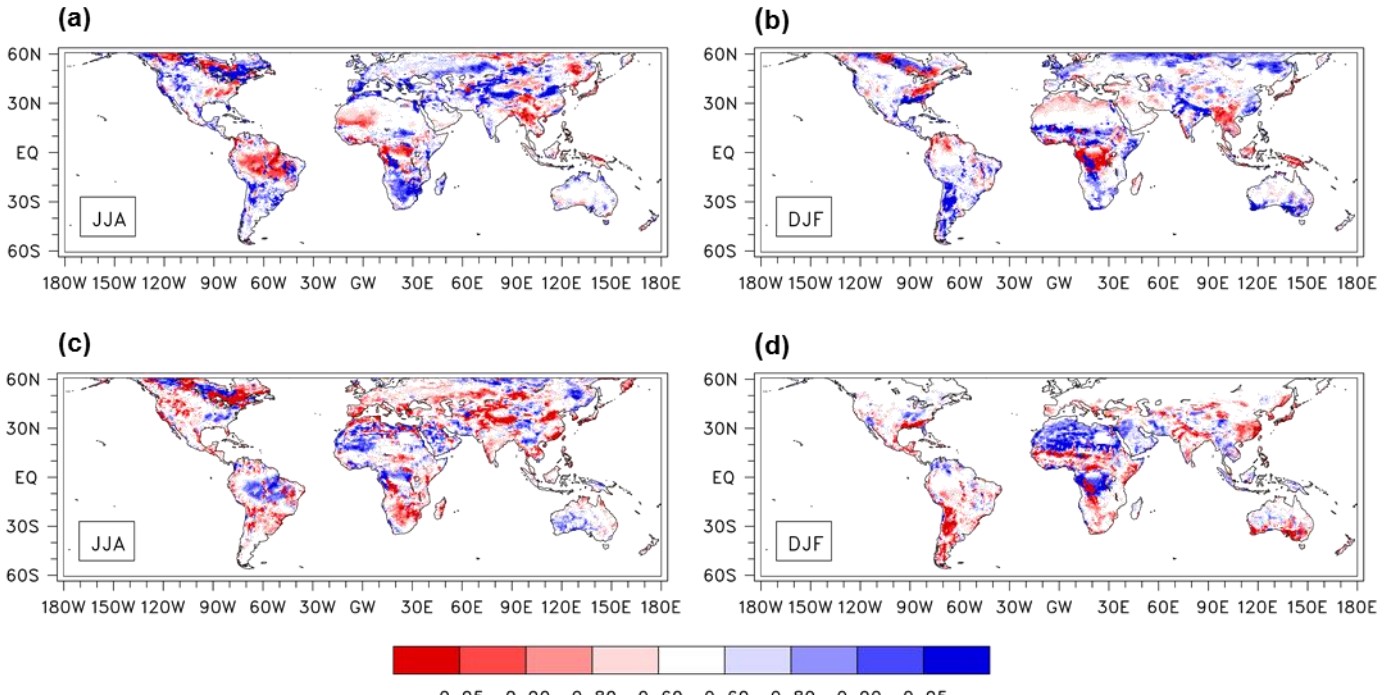

**Figure 6: Correlations (non-centred and no lag) between the latent heat flux and the soil moisture content (upper row) and between the sensible heat flux and the soil moisture content (lower row) in (a, c) boreal summer (JJA) and (b, d) austral summer (DJF). Units are standard unit; the indicated contours are at ±0.6, ±0.8, ±0.9 and ±0.95, respectively.**





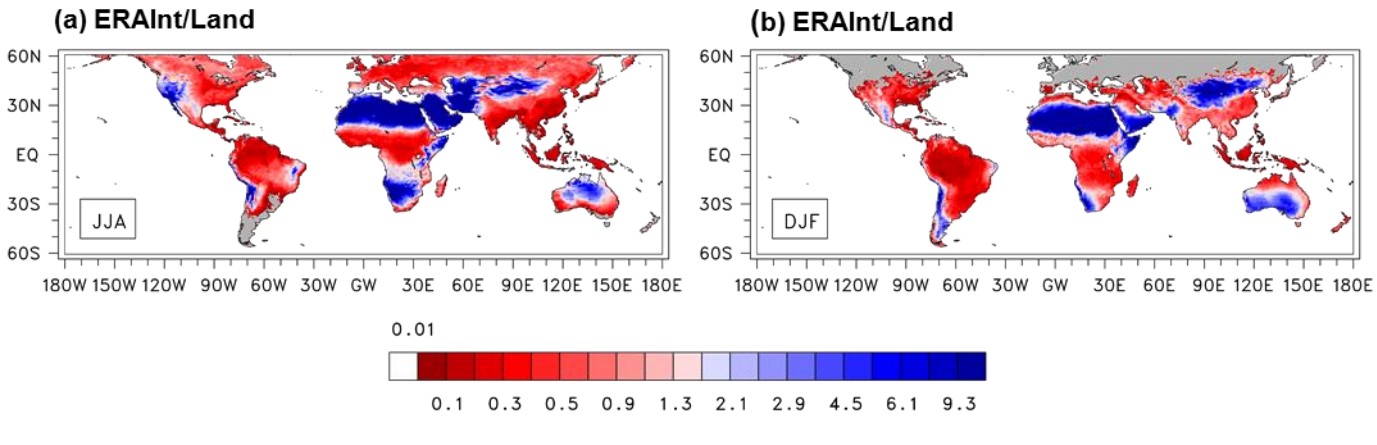

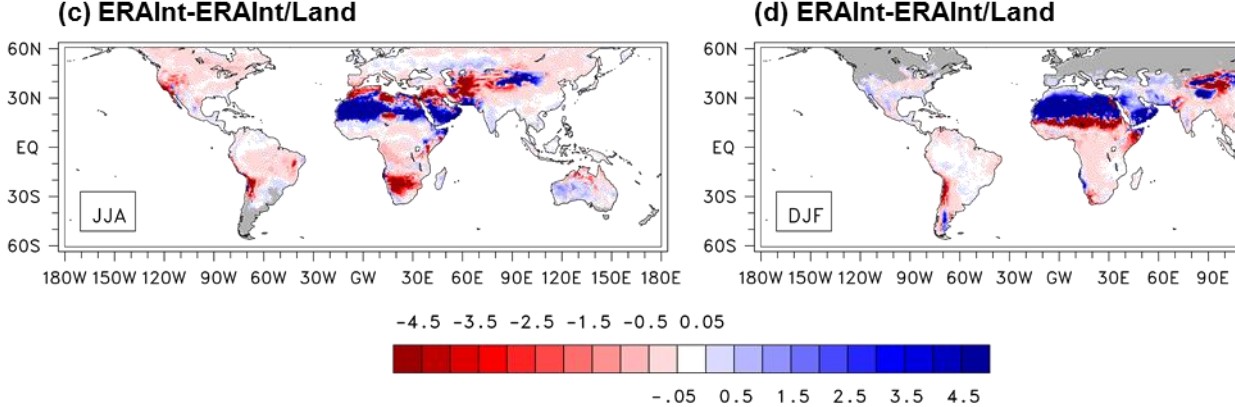

**Figure 7: As Fig. 4 but for the Bowen ratio (the ratio between the sensible and the latent heat flux). Units are standard unit; the contour intervals are varying between 0.1 and 1.6 (a, b) and 0.5 (c, d), respectively. Marked in grey are the areas, where the Bowen ratio is negative in the respective season.**





**Figure 8: As Fig. 4 but for the evaporative fraction (the ratio between the latent heat flux and the total surface energy flux). Units are 1/10 of the standard unit; the contour interval is 0.5. Marked in grey are the areas, where the evaporative fraction is negative in the respective season.**





**Bowen ratio − latent heat flux − sensible heat flux**

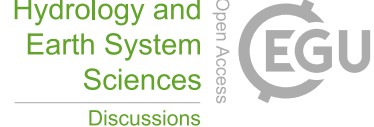



**Figure 9: Seasonal mean Bowen ratio, latent heat flux and sensible heat flux averaged over four regions in 12 seasons for ERAInt/Land (left column) and the corresponding differences between ERAInt and ERAInt/Land (right column). Units are W/m² for the surface energy fluxes (left axis) and 1/10 of the standard unit for the Bowen ratio (right axis), respectively. For the Sahel region (a, b), however, the unit is standard unit for the Bowen ratio (right axis).**

30





**Evaporative fraction – latent heat flux – sensible heat flux**





**Figure 10: Seasonal mean evaporative fraction, latent heat flux and sensible heat flux averaged over four regions in 12 seasons for ERAInt/Land (left column) and the corresponding differences between ERAInt and ERAInt/Land (right column). Units are W/m² for the surface energy fluxes (left axis) and 1/10 of the standard unit for the evaporative fraction (right axis), respectively.**

