# Peer review of "The role of improved soil moisture for the characteristics of surface energy fluxes in the ECMWF reanalyses"

_Hydrology and Earth System Sciences, 2017_

## Referee Comment (RC1) · Anonymous Referee #1 · 9 Jan 2018

Recommendation: minor revision

General comments

The study provides an assessment of the impact of improved land-surface scheme and precipitation on ECMWF reanalysis. Two reanalysis systems are compared: the standard ERA-Interim system and ERA-Interim/Land which includes adjusted precipitation and improved land-surface scheme. Both seasonal and regional effects of improved land-surface/precipitation are shown in results. The paper is well written and I recommend acceptance of the manuscript after a couple of specific comments and a few minor/technical points have been addressed.

Specific comments

1) ERAInt and ERAInt/Land differ not only for soil moisture treatment. Therefore the role of improved soil moisture cannot be separated from other differences in the two systems. Please discuss the limitations of the study.

2) Fig. 6: What is the significance level of the correlations?

Minor/technical comments

1) Title: I recommend to change the title to something like "The role of improved land-surface and precipitation for the characteristics of surface energy fluxes in the ECMWF reanalyses" See specific comment 1 above.

2) P1 L7: Change "more realistic" to "realistic"

3) P2 L4: Delete repetition "The coupling"

4) P2 L5: Change "soil moisture and precipitation and between soil moisture and temperature" to "soil moisture, precipitation and surface temperature"

5) P3 L26: Change "are related" to "are compared"

6) Table 1: domain boundaries are the same (10W-10E, 10N-20N) for WAF-S and WAF-C. Please check.

---

## Referee Comment (RC2) · Anonymous Referee #2 · 12 Jan 2018

This manuscript investigates the differences between ERA/Int and ERA/Int-Land seasonal mean regional soil moisture and surface fluxes, and attempts to link these back to the difference in precipitation between the two systems. While this topic is certainly of interest, the manuscript does not reveal any new findings and the author seems unaware of relevant literature and well established physical relationships.

MAJOR COMMENTS:

The main findings of the manuscript, as represented in the abstract and the conclusions, are i) soil moisture depends strongly on precipitation, and ii) soil moisture affects the partition of incoming radiation into sensible and latent heating, but does not affect the total incoming radiation. Both of these are basic features of land surface physics, and so do not represent a significant finding.

[Figure]

2. If the manuscript were to focus on relationships between different variables in the two systems, then these relationships would need to be shown quantitatively, rather than by qualitatively from comparing maps. Again, it is not enough to show that precipitation affects soil moisture, and soil moisture does not affect the incoming radiation. We know this.

3. All differences between ERA/Int and ERA/Int-Land are attributed to the different precipitation forcing, yet there are other major differences between the two systems. In particular, ERA/Int-Land includes a major update to the land model, ERA/Int-Land is a land-only replay while ERA/Int was coupled land/atmosphere system, and ERA/Int included a screen-level based land analysis which was not included in ERA/Int-Land. As such, the differences between the two systems cannot necessarily be attributed to the precipitation. The above differences need to be discussed, and the fact that the differences between ERA/Int and ERA/Int-Land output cannot be attributed to any one cause needs to be discussed in detail. For an example of how this issue could be addressed more thoroughly see:

Draper, C.S., R.H. Reichle, and R.D. Koster, Âǎ2018:ÂǎAssessment of MERRA-2 Land Surface Energy Flux Estimates.ÂǎJ. Climate,Âǎ31,Âǎ671–691,Âǎhttps://doi.org/10.1175/JCLI-D-17-0121.1Âǎ

SPECIFIC COMMENTS

P1, L18, and throughout the manuscript: the word 'diverges' is used, where 'differs' is more appropriate.

P1, L10: this is only one mechanism through which soil moisture can interact with the atmosphere. See Seneviratne et al (2010) for a discussion of this.

P2, L25: This is incorrect. GLACE used coupled land/atmosphere models, not offline simulations.

P2, L30: MERRA-2 has replaced MERRA, and should be referenced here.

P3, L5. Balsamo et al, 2015 also evaluated the land surface fluxes. Their results should be summarized here.

P4, L9: Three hours is the output resolution, not the "time step" or "temporal resolution".

P5, L 9: Limiting the domain to 60 degrees is an inefficient way to screen out snow and ice (in particular, there will be large snow covered areas equator-ward of 60 degrees).

Why not just screen these out directly? Also, does ERA/Int-Land use observed precipitation in the high latitudes? If not, this needs to be mentioned.

P5 last paragraph: Here it is assumed that the precipitation in ERA-Int/Land is necessarily more accurate than that in ERA/Int. While it is likely that the observation-informed precipitation in ERA-Int/land is overall more accurate, this cannot be assumed to always be the case. There are serious issues with the observed data sets too, particularly in regions that aren't well observed. This needs to be re-written to not assume that ERA/Int-Land is the truth, ie, ERA/Int should be described as being "lower than ERA/Int-Land", not as underestimating the precipitation / soil moisture".

P6 second paragraph: same comment as above. Here Africa and the Amazon are highlighted as having relatively large precipitation differences between the two systems. These regions are both very poorly observed, and these differences could be due to errors in the precipitation observations. Re-write as suggested above, and acknowledge the additional uncertainty in these regions.

P6 - P7 paragraph: I can't figure out what the author is trying to say here.

P7, L7: the use of "seasons" for a three month moving average is confusing. Just call it a three month moving average.

P9, L2: define "non-centered correlations" . Is "seasonal" true seasons, or your 3 month moving average ?

P9, L6: "coupling" has a specific meaning, and is not the right word here.
P9, paragraph starting L5: which system is being discussed here?

P9: there is no need to present both EF and Bowen ratio. They contain the same information, only one should be shown.

P12: 20 -25. I can't make sense of this paragraph. ERA-Int is already a coupled land/atmosphere system.

P12, L30: "They may also have remote effects by altering sea level pressure and, by this, the large scale circulation patterns". This is a very broad assertion, and needs to be discussed fully, and cited. Otherwise, delete it .

Figure 1-2: The color bars are saturated at the high end, so that any differences present are not evident. Please revise. You may want to use a non-linear scale for precipitation

It is not clear what is plotted in Figure 1f. The difference between the seasons, or between the reanalyses?

The labels on the top and bottom of the difference color bar don't match. Put all labels on the bottom.

Figure 6: is this for ERA/Int or ERA/Int-Land? Also include a better description of how the correlation is calculated (what exactly is the time series being used? A single JJA value for each year?)

---

## Author Comment (AC1) · 12 Jan 2018

Thank you for the comments. I will change the manuscript accordingly, once I have received the comments from the other referee(s).

In particular:

It is correct that not only the differences in soil moisture (mainly driven by precipitation) are responsible for the differences in the fluxes. I will revise the title and the text accordingly.

I will test the sigificance level of the correlations applying a suitable statistical technique, probably using a bootstrapping method.

---

## Referee Comment (RC3) · Anonymous Referee #3 · 15 Jan 2018

This paper presents a comparison of two successive ECMWF reanalysis datasets in terms of their surface hydrology. The paper lists a correction to match observational precipitation dataset as the main difference in forcing and presents resulting changes in global patterns of soil moisture and latent heat flux. While this may be useful as a reference to interpret other studies based on these same datasets, the discussion does not move far from a high-level qualitative assessment of features that can be explained by the model formulations.

**Major comment.**

The paper is full of spatial detail but seems to lack a clear definition of boundary conditions against which the results can be interpreted. As a consequence, it is not clearly formulated what assumptions are confirmed in this study or what unexpected features

of the model mechanics were discovered.

Specific comments:

1) Section 3. Differences in soil moisture and precipitation. I suggest starting the results with a presentation of the changes in precipitation between the two reanalysis datasets as this is a change in the forcing data. I expect there are earlier studies that present this too, which could be referenced. Are there other notable changes that may explain regional differences (e.g. soil parameterization, insolation, air temperature?). It is worth noting that the differences in mean soil moisture over desert areas do not originate from changes in precipitation (Fig 1e,f compared to Fig 2c,d). Can these be explained by changes in parameterization of the soil texture?

2) Section 4.1. The differences in latent heat versus sensible heat. First of all, please confirm that the total surface energy is not changed much between the two reanalysis sets. Second, because the models keep an energy balance, changes in one of these two terms are so obviously matched with a corresponding change in the other that it does not need to be treated as completely separate results. The description of sensible heat (page8, lines 18-31 and Figure 5) can be shortened accordingly to focus on the areas that do not conform this expectation (changes in available energy or ground heat flux).

3) Section 4.1. Correlation between differences in soil moisture and latent heat. Please expand on the clarification for negative correlations between differences in moisture and differences in latent heat. It is not clear if the authors explain this as the result of a chain of modeled interactions that ultimately leads to less evaporative demand, or that it simply is the result of parallel changes in the model formulation that are causing an apparent causality. The authors state ERAInt/land has the same meteorological forcing as ERAInt (page 3, line4) so I'm curious what pathway there is for increases in soil moisture to reduce latent heat. Also note that the caption of Figure 6 should mention that these correlations are between the differences, not the absolute values.

4) Regional analysis. Why are these particular regions selected?

5) I'm missing a true synthesis of the results where the main differences are summarized by climate region. Figures 3 and 10 could perhaps be combined for this purpose, if the 4 regions illustrate all the main differences: for example, increases and decreases in rainfall in water limited area, and increases and decreases in rainfall in energy limited areas.

6) General comment on presentation. The way differences are discussed in the paper make it a bit hard to follow at times. Two examples. 1) There is a lot of use of increases (decreases) linked to decreases (increases). Consider also using 'vice versa' or simply talk about the decreased variation or higher minimum or lower maximum values where that is appropriate. 2) Figure 3 could be easier to read if it had three panels on each row: soil moisture for both reanalysis, precipitation for each, and then the difference. This might show more clearly the change in seasonal variation going from one reanalysis to the next.